

# Effects of nitrogen topdressing and paclobutrazol at different stages on spike differentiation and yield of winter wheat

Dongxiao Li[1],[*], Shaojing Mo[1],[*], William D. Batchelor[2], Ruiting Cheng[1], Hongguang Wang[1] and Ruiqi Li[1]

[1] State Key Laboratory of North China Crop Improvement and Regulation/Key Laboratory of Crop Growth Regulation of Hebei Province/College of Agronomy, Hebei Agricultural University, Baoding, China
[2] Auburn University, Auburn, Alabama, United States of America
[*] These authors contributed equally to this work.

## ABSTRACT

**Background:** Optimal nitrogen (N) application and plant growth regulators can improve wheat productivity. This can help to improve yield level and ensure food security with limited resources in the Huang-Huai-Hai Plain of China (HPC).

**Methods:** A 2-year field experiment was conducted using a randomized block design with four treatments (TS-N topdressing at pseudostem erection stage ; TPS-N topdressing combined with paclobutrazol application at pseudostem erection stage; TJ-N topdressing at jointing stage; TPJ-N topdressing at combined with paclobutrazol application at jointing stage) in 2011–2013.

**Results:** The grain number per ear, thousand kernel weight and yield for the TJ and TPJ treatments were higher than those of the TS and TPS treatments. Grain number per ear, yield, and thousands kernel weigh for the TPJ treatment were significantly higher than for the TS and TPS in 2011–2012 (9.82% and 7.27%, 10.23% and 8.99%, 6.12% and 5.58%) and in 2012–2013 (10.21% and 11.55%, 8.00% and 6.58%, 0.00 and 0.00), respectively. Thousands kernel weight under TJ were significantly higher than those under TS and TPS by 13.21% and 14.03%, respectively in 2012–2013. The floret number, significantly correlated with cytokinin content, was also significantly increased under TJ and TPJ at connectivum differentiation stage. For TPJ treatment, the floret number was significantly higher than for the TS, TPS, and TJ by 19.92%, 10.21%, 6.10% in 2011–2012; it was higher than for the TS and TPS by 28.06% and 29.61% in 2012–2013, respectively. The relative expression level of cytokinin oxidase/dehydrogenase gene (*TaCKX2.2*) was improved during flowering, when cytokinin content was at high level and was also inhibited by paclobutrazol with different degrees.

**Conclusions:** Therefore, nitrogen topdressing at jointing stage had increased grain number per ear, thousand kernel weight, and grain yield of wheat. Paclobutrazol could delay spike differentiation and promote cytokinin accumulation that induced expression of *TaCKX2.2*, maintaining hormonal balance and affecting wheat spike morphogenesis.

Corresponding author
Ruiqi Li, wheatlrq@163.com

## INTRODUCTION

The North China Plain is one of the most important grain production regions in China and is experiencing conflicts between limited natural resources and crop production. Excessive use of nitrogen fertilizer is common in order to increase yields in the wheat-maize planting system. This often leads to large losses of nitrogen, resulting in serious environmental pollution (*Azam et al., 2009*; *Wang et al., 2010*). The planting area of winter wheat has been decreasing in order to enhance ecological benefits and due to limited water available for irrigated due to historic over-exploitation of ground water (*Xu et al., 2005*). In order to achieve China's food security goals, it is important to improve wheat yields. *Wang et al. (2015)* found that a key factor for increasing wheat yield was increasing the grain number per spike and seed weight, which is influenced by nitrogen fertilizer in the high-yield wheat region of Hebei Province. Nitrogen fertilizer promotes allocation of assimilate and increases assimilation amount and contribution rate of post-anthesis photosynthate to grain (*Ma et al., 2008a*). There are three nitrogen absorption peaks of winter wheat including fall vegetative growth before winter, the jointing-booting stage, and the flowering-filling stage. The transport of carbon assimilates to grain was found to be more efficient with nitrogen topdressing at the jointing stage than at the flowering stage (*Zhai & Li, 2006*). However, the nitrogen application should be delayed until booting stage under super-high-yield conditions (*Kara & Uysal, 2009*; *Liu et al., 2019*). The time of optimal nitrogen application is often related to local water conditions and wheat varieties. Under drought conditions or limited irrigation, nitrogen topdressing applied at the jointing stage was more beneficial to transport carbon photosynthate to grain compared to the flowering stage for winter wheat (*Gevrek & Atasoy, 2012*). However, for spring wheat, it was more beneficial to apply topdressing nitrogen at the flowering stage (*Walsh, Shafian & Christiaens, 2018*). Under irrigated conditions, a later topdressing application enhanced photosynthesis and increased wheat yield (*Chen et al., 2008*).

Paclobutrazol (pp333), a plant growth regulator that inhibits synthesis of endogenous gibberellin, is easily absorbed by wheat root, stem, and leaf. The application of paclobutrazol can inhibit crop height, increase stress tolerance, promote tillering, and increase spike number and yield (*Hajihashemi et al., 2007*; *Gómez et al., 2011*; *Peng et al., 2014*; *Dwivedi, Arora & Kumar, 2017*). Combining nitrogen topdressing with paclobutrazol can also improve photosynthetic rate, increase grain weight per spike, and achieve further increase in wheat yield when applied at the jointing stage (*Yang, Fan & Guo, 2008*). A combination of nitrogen topdressing at the pseudostem erection stage with a paclobutrazol application can increase dry matter accumulation, grain weight per spike, and grain yield (*Zhang et al., 2017*). The appropriate amount of nitrogen and paclobutrazol can increase stem lodging resistance, nitrogen uptake and maintain high duration of the flag leaf to promote high and stable yield (*Chen et al., 2011*; *Nouriyani et al., 2012a*, *2012b*). Paclobutrazol also regulated root morphological characteristics, maintained physiological function and root activity, increased grain yield by enhancing the levels of osmolytes, endogenous hormone contents, and antioxidant activities under

adverse environmental conditions (*Soumya, Kumar & Pal, 2017*; *Kamran et al., 2018*). Additionally, nitrogen can also regulate endogenous hormone content and affect ear and flower development (*Ma et al., 2008a*). Spikelet number, regulated by hormones in spike and roots, was closely related to nitrogen supply (*Chen et al., 2008*; *Zhang et al., 2009*). At the early grain formation stage, zeatin riboside (ZR) content could effectively regulate panicle flower development and spikelet number (*Ma et al., 2008a*). Decreasing free cytokinin (mainly zeatin and zeatin riboside) and ABA contents in grain and root would be possible resulting in sterile spikelet (*Chen et al., 2008*; *Zhang et al., 2009*). Paclobutrazol application could influence gibberellins (GAs) directly and modify flowering and development of plants (*Zhang et al., 2016*).

During ear differentiation, cytokinins oxidase/dehydrogenase (CKX), controlling endogenous cytokinins content, often negatively regulates time of flowering and grain formation of crops (*Zalewski et al., 2010*; *Yeh et al., 2015*; *Ashikari et al., 2005*; *Bartrina et al., 2011*). For wheat, *TaCKX2.2* was involved in the formation of the spike grain number and yield (*Zhang et al., 2011*). Inhibiting *TaCKX2* expression by RNAi increased the grain number per spike in bread wheat plants (*Li et al., 2018*). Previous studies have shown that the optimal application amount of a nitrogen and paclobutrazol combination at the stem elongation stage (*Nouriyani et al., 2012a*, *2012b*) and nitrogen topdressing at the jointing stage was advantageous for yield and grain quality (*Wu et al., 2014*). It is not clear what the optimum topdressing nitrogen combined paclobutrazol application is for optimal ear differentiation and individual wheat productivity, nor how changing phytohormones and related gene expression occur in plants under different nitrogen topdressing treatments in the North China Plain.

Our hypothesis was that nitrogen topdressing and paclobutrazol application at different stages would influence cytokinin (IPA and ZR) content of spike differentiation. Accordingly, the related gene expression varied. Combining molecular biology methods and crop growth analysis, we explored the differential expression law of cytokinin synthesis related genes in individual spike differentiation that directly influence grain yield. The objective of this study was to (1) evaluate the effect of topdressing nitrogen with paclobutrazol application on spike differentiation, and (2) elucidate the possible mechanisms of improving the productivity of wheat. The focus of this study was to determine the impact of N and paclobutrazol on individual spikelet formation, resulting differences in final yield, and to determine how phytohormones and related gene expression changes during spike differentiation. This study will help to provide guidance for optimizing fertilizer and increasing high yield potential of wheat in the North China Region.

## MATERIALS AND METHODS

### Experimental site and soil

A field experiment was carried out from 2011 to 2013 on super-high-yield testing farmland in Gaocheng, Shijiazhuang (37°79′N, 115°31′E), China. The climate was classified as subhumid continental monsoon. The mean annual rainfall is typically less than 520 mm,
**Table 1 Soil conditions of the experimental field.** Each data indicates the nutritious elements contents in different soil layers.

| Year | Soil layers (cm) | Organic matter (g·kg$^{-1}$) | Total N (g·kg$^{-1}$) | Available N (mg·kg$^{-1}$) | Available P (mg·kg$^{-1}$) | Available K (mg·kg$^{-1}$) |
|---|---|---|---|---|---|---|
| 2011–2012 | 0–20 | 13.2 | 1.1 | 148.8 | 33.0 | 128.6 |
| | 20–40 | 5.6 | 0.6 | 60.5 | 7.3 | 50.0 |
| 2012–2013 | 0–20 | 20.3 | 0.9 | 124.5 | 21.4 | 133.3 |
| | 20–40 | 9.6 | 0.4 | 42.0 | 7.5 | 52.6 |

one-third of which typically falls during the wheat-growing season. Winter wheat-summer maize rotation is the typical planting system in this region. The soil type was light-loamy Chao Soil and the soil properties are shown in Table 1.

### Experimental design and crop management

The experiment was a randomized complete block design with four experimental treatments: (1) nitrogen topdressing at pseudostem erection stage (TS), (2) TS with paclobutrazol application at pseudostem erection stage (TPS), (3) nitrogen topdressing at the jointing stage (TJ), and (4) TJ with paclobutrazol application at the jointing stage (TPJ). Each treatment was replicated 3 times. Each plot area was 60 m$^2$.

The cultivar Shimai18 was provided by Shijiazhuang Academy of Agricultural Sciences, Shijiazhuang, Hebei. This variety was planted on October 7, 2011 and October 9, 2012 with a row spacing of 0.15 m and plant density of $270 \times 10^4$ plants ha$^{-1}$. Fertilizers were applied before planting at rates of 112 kg ha$^{-1}$ nitrogen, 60 kg ha$^{-1}$ phosphorus, 87 kg ha$^{-1}$ potassium and 9 kg ha$^{-1}$ zinc. Topdressed N was applied at either pseudostem erection stage or jointing at a rate of 112 kg ha$^{-1}$. Fore-rotating corn straw from the previous crop was crushed 2 times and returned to the field at a 15 cm depth. Paclobutrazol (provided by Shenggong Bioengineering (Shanghai) Co., Ltd., Shanghai, China) was applied by a foliar spray of aqueous solution containing 10% wettable powder at an effective concentration of 200 mg kg$^{-1}$ which equated to a rate of 675 kg ha$^{-1}$ at either pseudostem erection or jointing stage.

### Variable measurements

Young ears were sampled and then quickly-frozen at −80 °C in the differentiation stages of stamen and pistils, early connectivum, late connectivum, and tetrad formation, separately. Partial young ears marked on the same flowering date were sampled at 0, 3, 6, 9 and 12 days after flowering. Beginning with the regreening stage, representative wheat plants were selected at 3 days intervals. The numbers of spikelets and florets were recorded in a constant state (on about 21$^{st}$ April) using an anatomical lens (Stemi 2000-c; Carl Zeiss, Jena, Germany).

Total ribonucleic acid (RNA) was extracted from wheat samples with a PlantRNA Kit (Tiangen, Beijing, China) according to the manufacturer's protocol. RNA concentration and purity were detected by spectrophotometer (Nanodrop, Wilmington, DE, USA); the RNA integrity was detected by agarose gel electrophoresis. cDNA obtained by reverse transcription of RNA was used for subsequent fluorescence quantitative expression.

First-strand cDNA was generated from 2 μg of total RNA using a PrimeScript II First Strand cDNA Synthesis Kit (TaKaRa, Dalian, China). The primer used for qRT-PCR were designed according to the published sequence *TaCKX2* (geneID: GU084177.1) by software Primer 5 (*Zhang et al., 2011*). Priming sequence: GGGAGAAGAAGCACTTTGGTC (*TaCKX2.2*-F); CCTGCAGTAAACTCAAACCATATC (*TaCKX2.2*-R). The optimal reference gene *GAPDH* (GAPDH-F: TGTCCATGCCATGACTGCAA; GAPDH-R: CCAGTGCTGCTTGGAATGATG) was selected for mRNA transcription studies using quantitative real-time polymerase chain reaction (PCR). All PCR reactions were repeated three times and the data were normalized to constitutively expressed TaGAPDH according to the 2−ΔΔCt method described by *Livak & Schmittgen (2001)*. Thermal-cycling conditions included an initial denaturation at 95 °C for 5 s, followed by 45 cycles at 95 °C for 5 s, 60 °C for 10 s, and 72 °C for 15 s, then a final melt step from 60 °C to 95 °C.

Endogenous phytohormone content (IAA, auxin; ABA, abscisic acid; GA, gibberellin; cytokinin, CTK including isopentenyl adenosine-IPA and trans zein nucleoside-ZR) was measured by enzyme linked immunosorbent assay (ELISA) (*Li et al., 2017*). The test kit was provided by College of Agriculture and Biotechnology, China Agricultural University, Beijing, China. Young ear samples were collected from each treatment and cut into 0.5 g pieces and were ground to a fine powder in liquid nitrogen with a pre-cooled mortar. The powder was diluted with extracting solution and was extracted over 4 h at 4 °C in cold 80% methanol based on the protocol outlined by *Oliver, Dennis & Dolferus (2007)*. The homogenate was then centrifuged at 3,500 rpm min⁻¹ for 20 min, and the supernatant was collected and washed by 80% methanol in a C-18 solid phase extraction column. Next, the filtered fluid of all the samples was pooled and dried in a vacuum chamber. Sample diluent was added to a final volume of 1 mL, which was diluted in tris-buffered saline and GA, CTK, ABA, IAA concentrations were measured according to *Jin et al. (2011)*.

Plant height was measured at pseudostem erection, jointing, booting, and ripening stages, respectively. Yield and yield components were measured by conventional methods. The harvest area was 3 m² for each plot. All plants were harvested, threshed, and dried to weigh seed yield. And 1,000-kernel weight was determined using the sum of two 500-kernel sample weights according with absolute value of the difference between two 500-kernel weights divided the average of both was less than 5%. If not, the third 500-kernel sample was weighted. The number of spikes was measured by counting the samples in one-meter-length of row in each replication at maturity. The number of kernels per spike was measured by counting the number of kernels per spike in 20 plants per plot.

## Statistical analysis

All data were run using analysis of variance (ANOVA) with three replicates according to Excel 2003, SPSS 17.0 (SPSS Inc., Chicago, IL, USA). The Duncan's new Multiple Range (DMR) test at 5% probability level was used to test the differences among the mean values. Significant differences were labelled based on DMR.

**Table 2 Plant height of wheat under different treatments.** Each data point indicates the average plant height of five repeats plants at different treatments.

| Year | Treatment | Plant height (cm) | | | |
|---|---|---|---|---|---|
| | | Pseudostem erection stage | Jointing stage | Booting stage | Maturity stage |
| 2011–2012 | TS | 17.8a | 29.7a | 60.3a | 69.2a |
| | TPS | 17.5a | 28.1b | 58.6b | 67.6b |
| | TJ | 19.0a | 27.6c | 57.1bc | 66.4b |
| | TPJ | 18.3a | 27.6c | 56.3c | 64.1c |
| 2012–2013 | TS | 19.1a | 27.1a | 54.4a | 68.7a |
| | TPS | 18.7a | 23.6b | 51.3b | 67.1b |
| | TJ | 19.0a | 26.0a | 50.7b | 66.0b |
| | TPJ | 19.4a | 26.0a | 48.1c | 61.7c |

Note:
Different letters in each column indicate significant differences among four treatments, assessed by ANOVA ($P \leq 0.05$). TS, nitrogen topdressing at erecting stage; TPS, TS combined with paclobutrazol application; TJ, nitrogen topdressing at jointing stage; TPJ, TJ combined with paclobutrazol application.

## RESULTS

### Plant height

Plant height increased gradually with the growth and development of seedlings in all treatments (Table 2). At pseudostem erection stage, there were no significant differences in plant height across all treatments. At the jointing stage, plant height in the TS treatment was significantly higher than in the TPS, TJ (2011–2012) and TPJ treatments (2011–2012). Plant height in the TJ and TPJ treatments were not significantly different. At booting stage, the plant height in the TS treatment was significantly higher than other treatments, while the TPJ treatment had significantly lower plant height than other treatments (2012–2013). No significant differences were existed between TPS and TJ. At mature stage, plant height reached to a peak value for all treatments. The paclobutrazol treatments had a significantly lower plant height than the non-paclobutrazol treatments.

### Wheat yield and yield components

Data from both years showed that grains per spike, thousand grain weight, and yield of wheat in the TJ and TPJ treatments were higher than those in the TS and TPS treatments (Table 3). The decreasing order of observed yield was TPJ>TJ>TPS>TS, and the TPJ yield was significantly higher than the yield for the TPS (10.22% and 8.88% higher) and TS (8.99% and 6.58% higher) treatments in 2011–2012 and 2012–2013, respectively. The topdressing N + paclobutrazol treatment yield components were not significantly different than the N treatments alone for the different paclobutrazol application dates. The thousand grain weight for the TJ and TPJ treatments were significantly higher than for the TS by 3.57% and 6.12%, respectively, and TPJ was significantly higher than TPS by 5.58% in 2011–2012. Thousand grain weight in the TJ treatment was significantly higher than in the TS and TPS treatments by 13.21% and 14.03%, respectively, in 2012–2013. This result suggested that nitrogen topdressing at the jointing stage was more

**Table 3 Wheat yield and yield components under different treatments.** Each data point indicates the wheat yield and yield components in three repeats testing plots with different treatments.

| Year | Treatments | Spikes ($\times10^4 \cdot ha^{-1}$) | Grains Per spike | Thousand grain weight (g) | Yield ($t \cdot ha^{-1}$) |
|---|---|---|---|---|---|
| 2011–2012 | TS | 796.7a | 33.6b | 39.2c | 8.8b |
| | TPS | 752.3a | 34.4b | 39.4bc | 8.9b |
| | TJ | 798.9a | 36.2a | 40.6ab | 9.4ab |
| | TPJ | 807.8a | 36.9a | 41.6a | 9.7a |
| 2012–2013 | TS | 829.4a | 33.3bc | 28.0b | 7.5b |
| | TPS | 834.9a | 32.9c | 27.8b | 7.6b |
| | TJ | 806.0a | 35.8ab | 31.7a | 7.8ab |
| | TPJ | 823.4a | 36.7a | 30.5ab | 8.1a |

Note:
Different letters in each column indicate significant differences among four treatments, assessed by ANOVA ($P \leq 0.05$). TS, nitrogen topdressing at erecting stage; TPS, TS combined with paclobutrazol application; TJ, nitrogen topdressing at jointing stage; TPJ, TJ combined with paclobutrazol application.

**Table 4 Spike differentiation of wheat under different treatments.** Each data point indicates the average performance of five repeats in spike differentiation indexes under different treatments.

| Year | Treatments | Differentiation Spikelet number | Differentiation Floret number | Floret number differentiated from stamen and gynoecium | Floret number differentiated from connectivum |
|---|---|---|---|---|---|
| 2011-2012 | TS | 20.8a | 140.8b | 118.4b | 52.2c |
| | TPS | 19.2b | 138.2b | 102.2c | 56.8b |
| | TJ | 20.8a | 145.4ab | 115.4b | 59.0b |
| | TPJ | 20.4ab | 150.2a | 123.6a | 62.6a |
| 2012–2013 | TS | 19.4a | 148.8a | 99.4b | 67.0b |
| | TPS | 18.4b | 141.8b | 96.4b | 66.2b |
| | TJ | 18.8ab | 150.4a | 112.6a | 87.6a |
| | TPJ | 19.0ab | 149.6a | 113.8a | 85.8a |

Note:
Different letters in each column indicate significant differences among four treatments, assessed by ANOVA ($P \leq 0.05$). TS, nitrogen topdressing at erecting stage; TPS, TS combined with paclobutrazol application; TJ, nitrogen topdressing at jointing stage; TPJ, TJ combined with paclobutrazol application.

beneficial to grain weight than at the pseudostem erection stage. The grains per spike in TJ and TPJ were significantly higher than in TS and TPS in 2011–2012, while grains per spike in TPJ was significantly higher than in TS and TPS in 2012–2013. Grains per spike for the TJ treatment was significantly higher than that of TPS in 2012–2013. There were no significant differences in spikes per hm² among four treatments.

## Spike differentiation

The numbers of spikelets and florets in the TPS treatment were less than those in other treatments (Table 4). Spikelet number in the TS treatment, the most one, was significantly higher than in the TPS treatment by 8.33% and 5.43% in 2011–2012 and 2012–2013, respectively. But spikelet number in the TS had no obvious differences with that in the TJ and TPJ treatments. Florets number in the TPJ was significantly higher than in the TPS and TS treatments by 8.68% and 6.68% in 2011–2012; and florets number in the TPS

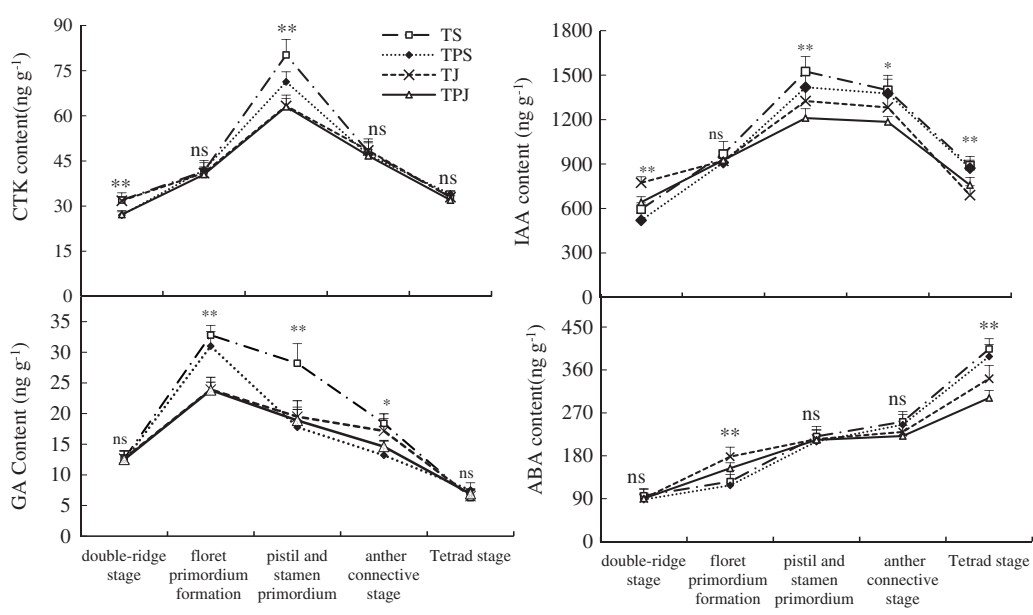

**Figure 1 Dynamic of endogenous hormone levels in ears during spike differentiation.** Each data point indicates the average performance of five repeats in phytohormone contents with different stages. Asterisks (*, **) indicate significance at the 0.05 and 0.01 probability levels, respectively; ns, not significant.

treatment was significantly lower than in the other treatments in 2012–2013. The number of florets differentiated from stamen and pistil of wheat in the TPJ treatment was significantly higher than in the TJ (2011–2012, 7.11% higher), TS (4.39% and 14.49% higher), and TPS (20.94% and 18.05% higher) treatments, respectively; this number in the TJ was significantly higher than in the TPS and TS treatments by 13.28% and 16.81% in 2012-2013, respectively. The floret number at the connectivum differentiation stage in the TPJ treatment was significantly higher than in the TS, TPS, and TJ treatments by 19.92%, 10.21%, and 6.10%, respectively, in 2011–2012; and the numbers in the TJ and TPJ were significantly higher than in the TS (30.75% and 28.06%) and TPS (32.33% and 29.61%) treatments in 2012–2013, respectively. These results suggested that N topdressing and paclobutrazol application at jointing stage was more helpful in improving development of florets, inhibiting its degeneration in wheat, and increasing grains number per spike.

## Phytohormone content

The variation of endogenous hormone levels in wheat spikes were similar among all treatments during differentiation of the main stem spike (Fig. 1). However, the content of endogenous hormones was different at various phases of spike differentiation. The changes in CTK, IAA, and GA content showed a single-peak curve. CTK contents in the TS and TJ treatments was significantly higher than in the TPS (18.52% and 17.78%) and TPJ (17.65% and 16.91%), respectively, at the double-ridge stage. The peak value of CTK content in the TS treatments was significantly higher than in the TPS treatment by 12.48% at the stamen and pistil differentiated stage, when CTK content in the TPS was
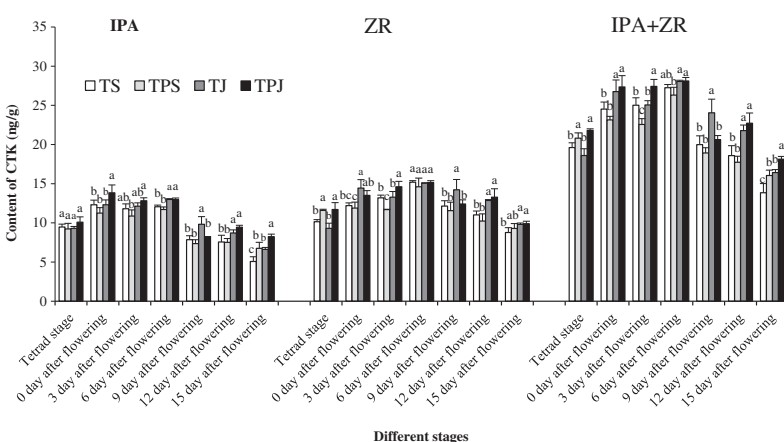

**Figure 2 IPA and ZR content of spikes under all treatments at different stages.** Each data point indicates the average performance of five repeats in spike with different treatments during all flowering stage. Different letters indicate significant differences among four treatments, assessed by ANOVA ($P \leq$ 0.05).

significantly higher than in the TJ and TPJ by 12.64% and 13.17%, respectively. The peak values of GA in the TS and TPS were significantly higher than in the TJ (37.12% and 29.60%) and TPJ (37.64% and 30.09%) treatments, respectively, at the floret primordia differentiation stage. IAA content in the TS and TPS treatments were significantly higher than in the TJ (15.04% and 7.02%) and TPJ (25.98% and 17.19%) treatments, respectively, at the pistil and stamen differentiated stage. These results showed that the nitrogen topdressing at the pseudostem stage promoted accumulation of growth acceleration hormones more than at the jointing stage. The differences between TS and TPS and between TJ and TPJ were increasingly smaller during the panicle differentiation stage. ABA content in spikes showed an increasing trend throughout the growth period. At the tetrad stage, ABA reached the maximum value and showed a descending order of TS>TPS>TJ>TPJ; ABA contents in the TS and TPS treatments were significantly higher than in the TJ (18.22% and 13.69%) and TPJ (34.07% and 28.93%) treatments, respectively. These results suggested that ABA content could be decreased under nitrogen topdressing with paclobutrazol application at the jointing stage.

## IPA and ZR content

Figure 2 shows single-peak trend in IPA, ZR, and (IPA+ZR) content in wheat grains under all treatments from tetrad stage to flowering stage. The peak values appeared mainly at 0–6 days after flowering. The IPA content, which was the highest in the TPJ treatment at 0, 3, 12, and 15 days after flowering, was significantly higher in the TJ and TPJ treatments than in the TS and TPS treatments by 2.71% and 8.51%, 11.54% and 17.83%, respectively, at 6 days after flowers. The ZR contents in grains for the TPS and TPJ treatments were both significantly higher than for the TS (14.15% and 15.41%) and TJ (24.54% and 25.91%) treatments, respectively, at tetrad stage. The ZR contents were significantly higher in the TPJ and TJ treatments than in the TS (20.43% and 16.62%) and TPS (29.75% and 25.64%) treatments, respectively, at 12 days after flowering. Additionally,

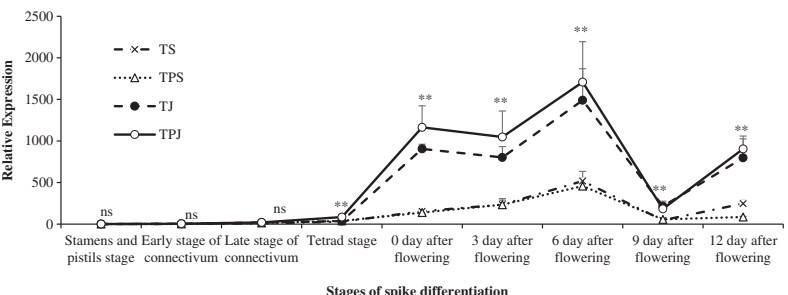

**Figure 3 The changing trend of relative expression of TACKX2.2 at different stages.** Each data point indicates the average performance of three repeats samples after randomly choosing with all treatments at different spike differentiational stages. Asterisks (\*, \*\*) indicate significance at the 0.05 and 0.01 probability levels, respectively; ns, not significant.

the ZR contents in the TPJ and TJ treatments were significantly higher than the others treatments, respectively, at 3 and 9 days after flowering. The IPA+ZR content showed a similar changing trend to ZR content and was higher in the TJ and TPJ treatments than in the TS and TPS treatments. The IPA+ZR content in the TPJ treatment was significantly higher than others at 3 and 15 days after flowering.

## *TaCKX2.2* expression

The trend of *TaCKX2.2* expression is shown in Fig. 3 under different treatments during spike development. From the stamen and pistil differentiation to tetrad differentiation stage, the expression level of *TaCKX2.2* was very low in all treatments. The level rose sharply and reached a peak on 6 days after flowering under the TS and TPS treatments, which were significantly lower than the TJ (65.22% and 69.27%) and TPJ (69.67% and 73.21%) treatments, respectively. From 0–12 days after flowering, the *TaCKX2.2* expression in spikes under the TJ and TPJ treatments was significantly higher than that under the TS and TPS treatments, respectively. The first peak for the TJ and TPJ treatments occurred on 0 days and the second higher peak occurred on 6 days after flowering. At 9 days after flowering, *TaCKX2.2* expression was decreased substantially in all treatments. These results suggested that *TaCKX2.2* expression could be induced by paclobutrazol at the jointing stage and its expression in TPJ reached a maximum value during the flowering stage.

## Correlation analysis

The floret number had a significant linear correlation with IPA and ZR content from 0–12 days after flowering (Table 5). At 0, 6, and 12 days after flowering, the florets numbers from the stamen and gynoecium and from connectivum were all very significantly correlated to cytokinin (IPA+ZR) content. At 3, 9, and 15 days after flowering, there was a significant correlation between cytokinin (IPA+ZR) content and numbers of florets differentiated from stamen and gynoecium (15 days exception) and from connectivum. These results showed that both IPA and ZR could control and regulate floret number and development during full-blossom period.

**Table 5 The correlation between (IPA+ZR) content and floret number.** Each data point indicates the correlation coefficient of 12 runs between (IPA+ZR) content and floret number differentiated from connectivum, stamens and pistils.

| Content of (IPA+ZR) | Florets differentiated from connectivum | Florets differentiated from stamen and gynoecium |
|---|---|---|
| 0 day after flowering | 0.825** | 0.855** |
| 3 days after flowering | 0.685* | 0.63* |
| 6 days after flowering | 0.766** | 0.725** |
| 9 days after flowering | 0.600* | 0.683* |
| 12 days after flowering | 0.913** | 0.875** |
| 15 days after flowering | 0.695* | 0.549 |

**Note:**
*, **Significant at the 0.05 and 0.01 probability levels, respectively, ns, no significant. Each data point represents the average of the measured data.

## DISCUSSION

It has been reported that nitrogen topdressing at the jointing stage can increase the number of grains per ear and wheat yield (*Li et al., 2010*). The application of paclobutrazol at the stem elongation stage can promote nitrogen uptake and reduce nitrogen leaching to improve nitrogen use efficiency and wheat yield (*Nouriyani et al., 2012b*). In this study, the number of grains per ear, thousand kernel weight, and yield of wheat in the TPJ treatment were significantly higher than those in the TS and TPS treatments, which was in agreement with the report of *Wu et al. (2014)*. Paclobutrazol application decreased plant height from booting stage to maturity stage, and further reduced plant height when the application time was delayed from the pseudostem erection stage to the jointing stage. The inhibitory effect of paclobutrazol on overgrowth of wheat plants likely reduced excess consumption of nutrients and reserved photosynthate accumulation to support grain growth (*Ghosh et al., 2010*). Combing nitrogen topdressing and paclobutrazol application at jointing stage increased wheat yield. Although a single application of paclobutrazol had no significant effect on yield and yield components, using topdressing nitrogen with paclobutrazol gave higher yield component values than treatments without the paclobutrazol application. It was reported that the best effect was achieved using a combination of 150 mg $L^{-1}$ of paclobutrazol and 160 kg $ha^{-1}$ nitrogen, for which paclobutrazol increased the absorption and transportation of nitrogen in plant and significantly affected photosynthetic pigments to maintain a high duration of flag leaf area which increased the grain yield (*Nouriyani et al., 2012a*). Paclobutrazol was found to also reduce plant height and flag-leaf area of black rice, but increased sucrose and amylopectin content in the grain at a concentration of 50 ppm (*Dewi & Darussalam, 2018*). Previous reports indicated that paclobutrazol was prone to regulation on physiological balance at early growth stages, but not on final yield of winter wheat directly (*Yang, Fan & Guo, 2008*). Therefore, the regulatory effect of paclobutrazol is different for different developmental phase of plants and concentration variation from paclobutrazol or nitrogen or both.

The panicle primordium differentiation stage is a critical phase in reproductive organ construction and the formation of final yield of wheat. Nitrogen can increase floret survival rate and grain number in wheat (*Ferrante, Savin & Slafer, 2010*). Paclobutrazol can promote fertile florets differentiation and coordinate the vegetative and reproductive growth (*Hampton & Hebblethwaite, 2006*). Further observations have shown that the differentiation of young panicles was inhibited and the floret number was decreased under nitrogen topdressing with paclobutrazol at the pseudostem erection stage compared with the jointing stage. This may explain why grain number per ear and yield of wheat with nitrogen topdressing at the pseudostem erection stage was lower than at jointing stage. It is known that the growth of young ears and development of spikelets is closely related to hormone content (*Zhang et al., 2009*). A recent study reported that paclobutrazol elevated endogenous auxin and abscisic acid levels, suppressed gibberellins (GA4) and trans-zeatin concentrations of plant (*Opio et al., 2020*). However, another report showed that paclobutrazol significantly decreased the content of GA3 and IAA, and increased ABA contents in leaves of wheat (*Aly & Latif, 2011*). This study indicated that the paclobutrazol application decreased the number of spikelets and florets at the pseudostem erection stage, but increased floret number at the jointing stage based on 2-years data. The application of paclobutrazol inhibited not only GA content during spike differentiation but also CTK, IAA, and ABA content before the tetrad stage. This may be possible due to dynamic phytohormone levels and an unconstant inhibitory effect of paclobutrazol on spike differentiation. The mutual antagonism and interactions between those hormones varied in different growth stages of crops (*Wu et al., 2019*). Additionally, IAA, CTK, and GA contents in spikes were lower or not different under nitrogen topdressing with or without paclobutrazol at the jointing stage compared with the pseudostem erection stage. This result suggested that the differences in number of spikelets and florets among all treatments were not primarily determined by the phytohormone levels at this stage. At the tetrad stage, ABA content in spikelets under the TPS treatment was higher than that under the TPJ stage, suggesting that other promoting hormones likely regulated panicle and floret development after the tetrad stage. These results have also verified the hypothesis that individual spikelets number was affected by both nitrogen topdressing and paclobutrazol application, which played a promoting role mainly during the jointing stage.

Cytokinin, which is degraded by cytokinin oxidizes/dehydrogenase to IPA and ZR, plays an important role in early spikelet development and panicle differentiation (*Ma et al., 2008b*; *Jameson & Song, 2015*). From the tetrad to anthesis stage, cytokinin content in spikelets may be a regulatory mechanism. At the tetrad stage, ZR content in spikelets under the nitrogen topdressing with paclobutrazol treatment was significantly higher than that without the paclobutrazol application, which help to increase yield sink capacity (*Dewi & Darussalam, 2018*). From this stage onwards, the contents of ZR and IPA and the expression level of *TaCKX2.2* were higher for the nitrogen topdressing during 0–6 days after flowering. The increment was higher at the jointing stage than at the pseudostem

erection stage. It is likely that cytokinin oxidation/dehydrogenase would perform stronger degradation to maintain hormone balance when accumulation of cytokinin in ear exceeded demand, which often results from the rapid development of wheat ears with nitrogen topdressing during reproductive growth (*Werner et al., 2006*; *Zhang et al., 2011*). Panicle development was faster under nitrogen topdressing at the jointing stage, when *TaCKX2.2* expression was higher than that at the pseudostem erection stage. Meantime, paclobutrazol application increased *TaCKX2.2* expression to inhibit cytokinin content. Our study showed that the contents of IPA and ZR in the TPJ treatment, and ZR and (IPA+ZR) in the TPS treatment at 3 and 9 days after flowering, respectively, were suppressed by paclobutrazol, which was partly consistent with *Opio et al. (2020)*. However, the contents of ZR and ZR+IPA in TPJ were elevated significantly by paclobutrazol at 3 days after flowering. These results can be interpreted that paclobutrazol induced early flowering by increasing ABA and cytokinins contents in buds, which regulated increases in leaf water potential and carbon-nitrogen ratio of mango (*Upreti et al., 2013*). This study also showed that cytokinin (IPA+ZR) content was significantly correlated with the floret number at the stamen and gynoecium and the connectivum differentiation stage. It was established that paclobutrazol regulates floret differentiation and development indirectly by regulating cytokinin content and influencing the final yield. These results further verified the hypothesis that spikelet number and differentiation was regulated by phytohormone, in which cytokinin (IPA and ZR) content and *TaCKX2.2* expression could keep homeostasis mainly after the tetrad stage. Other factors such as water status, paclobutrazol concentrations, and growth stage should also be considered in understanding the regulatory process. Additionally, the differences in gene expression in spikes among treatments was not only influenced by cytokinin content but also by multiple hormone interactions and other hormone levels changing at crucial stages. Further research is needed to determine the primary molecular mechanisms by which assimilate of grain increased for the nitrogen and paclobutrazol treatment.

## CONCLUSION

In conclusion, nitrogen topdressing at jointing stage was optimal for increasing floret number, kernels number per spike, and yield. This result was partly related to cytokinin (especial ZR) content increasing after tetrad stage. Application of paclobutrazol could delay spike development to some extent and promote cytokinin content elevation, along with *TaCKX2.2* relative expression increasing, likely to keep hormonal equilibrium and regulate spikelet and floret number. This work has systematically combined the molecular biology, phytohormone physiology and crop growth to analyze seed yield increasing approach. Future research on the deep mechanism of interaction between paclobutrazol and phytohormone is needed. Therefore, our study shows still an effective measure to promote wheat productivity by regulating individual spikelet differentiation with nitrogen topdressing and paclobutrazol application at jointing stage. For conventional production wheat variety, there also need adequate planting population to achieve high yield and ensure food security in the North China Plain.

### Funding

This work was supported by the National System of Modern Agriculture Industrial Technology Project (CARS-03-05), the National Key Research and Development Program of China (2017YFD0300909), the Scientific Research Project of Hebei Education Department (QN2019046), the Hebei Province Natural Science Foundation for Youth (C2019204358), the Science and Technology Program of Baoding (1911ZN010), and the National Natural Science Foundation of China (No. 31871569). The funders had no role in study design, data collection and analysis, decision to publish, or preparation of the manuscript.

### Grant Disclosures

The following grant information was disclosed by the authors:
National System of Modern Agriculture Industrial Technology Project: CARS-03-05.
National Key Research and Development Program of China: 2017YFD0300909.
Scientific Research Project of Hebei Education Department: QN2019046.
Hebei Province Natural Science Foundation for Youth: C2019204358.
Science and Technology Program of Baoding: 1911ZN010.
National Natural Science Foundation of China: 31871569.

### Competing Interests

The authors declare that they have no competing interests.

### Author Contributions

- Dongxiao Li conceived and designed the experiments, performed the experiments, analyzed the data, prepared figures and/or tables, authored or reviewed drafts of the paper, and approved the final draft.
- Shaojing Mo performed the experiments, prepared figures and/or tables, and approved the final draft.
- William D. Batchelor analyzed the data, authored or reviewed drafts of the paper, and approved the final draft.
- Ruiting Cheng analyzed the data, authored or reviewed drafts of the paper, and approved the final draft.
- Hongguang Wang analyzed the data, prepared figures and/or tables, authored or reviewed drafts of the paper, and approved the final draft.
- Ruiqi Li conceived and designed the experiments, authored or reviewed drafts of the paper, and approved the final draft.

### Data Availability

The raw measurements are available in the Supplemental File.

## Supplemental Information

Supplemental information for this article can be found online at http://dx.doi.org/10.7717/peerj.12473#supplemental-information.

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
