# Peer review of "Effects of nitrogen topdressing and paclobutrazol at different stages on spike differentiation and yield of winter wheat"

_PeerJ, doi:10.7717/peerj.12473_

## Round 0.1 · original submission · Major Revisions

Please revise as per the reviewer's comments.

·

Basic reporting

OK

Experimental design

OK

Validity of the findings

OK

Regulation of nitrogen topdressing and paclobutrazol

Additional comments

(1) Primary research should be hypothesis driven. Any hypothesis, which authors had at the beginning of the study, has, unfortunately, not been outlined by the authors, for example at the end of the “Introduction”. This should also include outlining in the “Discussion” if authors actually found support for their original hypothesis after carrying out their research work.
(2) The objectives of this study were to investigate Regulation of nitrogen topdressing and paclobutrazol in winter wheat varieties . It would be useful for the reader to outline what particular research question(s) authors have asked to address the set objective.
(3) In the final “Conclusions” authors should briefly summarize the new results obtained by carrying out the research. In addition, authors should outline how their study has advanced our current knowledge on the importance of nitrogen topdressing and paclobutrazol in winter wheat varieties, and to understand how the regulation effect of the nitrogen topdressing and paclobutrazol in winter wheat varieties
(4) A final statement is also required on how the study has finally contributed to food security..

·

Basic reporting

This work presents interesting results on the regulation of nitrogen topdressing and paclobutrazol
at different stages on spike differentiation of winter wheat. In the present study, nitrogen topdressing and paclobutrazol significantly alleviated the yield in winter wheat by regulating the hormonal balance. Also explained well the hormonal regulation and expression TACKX2.2 gene and their role in reproductive organs development and differentiation.


However, there are few limitations.

There is a lot of grammar and English issues. The authors did not maintain the control plants (without nitrogen topdressing and paclobutrazol) to compare the absolute effect of the treatments on growth and yield. Material and methods need to provide more details especially, RNA extraction and gene expression analysis part. The paper is written very descriptively, especially the hormonal results part, with many explanations that make the manuscript difficult for non-specialists to understand. It needs a lot of improvement; simplify the writing style and it needs to be crisp. Besides, there are many uniformity issues in the references.

Experimental design

The experimental design is good but the authors did not maintain the control plants (without nitrogen topdressing and paclobutrazol) to compare the absolute effect of the treatments on growth and yield. Material and methods need to provide more details especially, RNA extraction and gene expression analysis part.

Validity of the findings

Introduction: Well written, but, in the introduction part, the study is missing the rationale, how this study differs from the previous studies (Wu et al. 2014, Nouriyani et al. 2012a and b), and the significance of proposed objectives.
Materials and methods need more details in few places.

Results: Well written except phytohormone content part. The authors need to rewrite that part.
Discussion: Well discussed the obtained results. However, it needs some improvement.
Conclusion: Well written

Additional comments

This work presents interesting results on regulation of nitrogen topdressing and paclobutrazol
at different stages on spike differentiation of winter wheat. In the present study, nitrogen topdressing and paclobutrazol significantly alleviated the yield in winter wheat by regulating the hormonal balance. Also explained well the hormonal regulation and expression TACKX2.2 gene and their role in reproductive organs development and differentiation.


However, there are few limitations.

There is a lot of grammar and English issues. The authors did not maintain the control plants (without nitrogen topdressing and paclobutrazol) to compare the absolute effect of the treatments on growth and yield. Material and methods need to provide more details especially, RNA extraction and gene expression analysis part. The paper is written very descriptively, especially the hormonal results part, with many explanations that makes the manuscript difficult for non-specialists to understand. It needs a lot of improvement; simplify the writing style and it needs to be crisp. Besides, there are many uniformity issues in the references.

Title: Maybe it needs to change. Title should be focused to the yield related.

Abstract: Good,
Line 17-18: Better to mention the percentage change in the brackets between, how significant the TPJ is compared to other treatments (TJ, TS and TPS).
Line 20: Correct the enzyme name... change it to cytokinin oxidase/dehydrogenase.

Introduction:

Well written, but, in the introduction part, the study is missing the rationale, how this study differs from the previous studies (Wu et al. 2014, Nouriyani et al. 2012a and b), and the significance of proposed objectives.
Line
Line 72: Correct the enzyme name... change it to cytokinin oxidase/dehydrogenase.
Line 82: Change improving productivity to improving the productivity.

Materials and methods:

Need more details in few places.
Line 88: Add the geographical coordinates for the field.
Line 93: Add the information. Where is it located?
Line 98: Add the information. From which company (like sigma) was this product brought? and the place details.
Line 99-100: Not clear rewrite the sentence by mentioning the doses correctly.
Line 101: Change the 270◊104 to 270×104
Line 113: What method was used for RNA extraction? Add the reference. If the kit method was used, add the product details (company, city and country).
Line 115: Add the information cDNA kit product details (company, city and country).
Line 116-117: Provide the primer source reference. If authors designed? add the which software used for the primer designing?
Line 118: Provide the reference gene sequence.
Line 119: How did the relative gene expression was calculated (2–∆∆Ct)? Provide the reference.
Also, provide the qPCR conditions (Program details).
Line 120-122: Cite the reference/ kit product details for the ELISA method.
Line 135: Would be.. will used to indicate the future so change the sentence. If not, the third 500-kernel sample was/were weighed. Correct it.
Results:
Well written except phytohormone content part. Authors need to rewrite that part.
Line 159: Change the sentence to.. Whether spraying paclobutrazol or not,
Line 167: Was that 1.81-3.81%? Correct it.
Line 187-189: Not clear what authors need to say here. Rewrite.
Line 196-197: This sentence is not clear. Talking about which hormone is increasing?
The differences between two paclobutrazol treatments were increasingly smaller during the panicle differentiation.
Line 208: Change flowers to flowering.
Line 215-226: Correct the gene name from TACKX2.2 to TaCKX2.2
Line 229 and 231: No need to write the day after each number. Change to.. 0,6, and .12 days after flowering.


Figure 1: Better to change the line format between TS, TPS and TJ, and TPJ to avoid confusion (like how they did for the figure 3 solid and dotted lines).

Discussion:

Well discussed the obtained results. However, it needs some improvement.
Need the explanation for the relationship between the plant height and yield. As the plant height was high on TS and TPS compared to TJ and TPJ, explain how plant height could impact the hormonal content, reproductive organ development, and yield using the previous literature?
Line 243-244: Authors did not measure any parameters related to photosynthesis or NUE, then, how can they say that nitrogen topdressing combined with paclobutrazol at the jointing stage promoted more photosynthate accumulation in grain and improved nitrogen use efficiency? Need an explanation.
Line 260: duo (or due) ..correct the spelling.
Line 262: Give the space.. in reproductive
Line 266: Using could be and obviously in the same sentence does not sound good. Rewrite the sentence.

Line 288 and 297: Correct the enzyme name... change it to cytokinin oxidase/dehydrogenase
Line 289: Italicize the author name.
Line 296: Change the suggested to suggesting.
Line 301 and 302: Line 215-226: Correct the gene name from TACKX2.2 to TaCKX2.2

Conclusion:
Well written
Delete this part… In this paper, it could be concluded that and replace it with … In conclusion,

Check the attached PDF for more corrections

---

## Round 0.2 · Minor Revisions

Please follow the comments on the annotated PDF provided by the Section Editor and revise.

---

## Round 0.3 · accepted · Accept

This manuscript has been revised according to the reviewer's comments. Now, it is acceptable for publication.